# Edible Coatings as a Natural Packaging System to Improve Fruit and Vegetable Shelf Life and Quality

**DOI:** 10.3390/foods12193570

**Published:** 2023-09-26

**Authors:** Ana Perez-Vazquez, Paula Barciela, Maria Carpena, Miguel A. Prieto

**Affiliations:** Nutrition and Bromatology Group, Department of Analytical Chemistry and Food Science, Faculty of Science, Universidade de Vigo, E32004 Ourense, Spain; anaperezvazquez00@gmail.com (A.P.-V.); pau_barci@hotmail.es (P.B.); mcarpena@uvigo.es (M.C.)

**Keywords:** edible coating, shelf life, biopolymers, food waste, fruits, vegetables, by-products

## Abstract

In the past years, consumers have increased their interest in buying healthier food products, rejecting those products with more additives and giving preference to the fresh ones. Moreover, the current environmental situation has made society more aware of the importance of reducing the production of plastic and food waste. In this way and considering the food industry’s need to reduce food spoilage along the food chain, edible coatings have been considered eco-friendly food packaging that can replace traditional plastic packaging, providing an improvement in the product’s shelf life. Edible coatings are thin layers applied straight onto the food material’s surface that are made of biopolymers that usually incorporate other elements, such as nanoparticles or essential oils, to improve their physicochemical properties. These materials must provide a barrier that can prevent the passage of water vapor and other gasses, microbial growth, moisture loss, and oxidation so shelf life can be extended. The aim of this review was to compile the current data available to give a global vision of the formulation process and the different ways to improve the characteristics of the coats applied to both fruits and vegetables. In this way, the suitability of compounds in by-products produced in the food industry chain were also considered for edible coating production.

## 1. Introduction

In recent decades, consumers have become more worried about their food habits, rejecting products with additives and giving preference to fresh ones [1]. Furthermore, society has increased its concern about the environment, leading to an increased interest in reducing plastic consumption and food waste. The food industry has been struggling with the loss of food quality and quantity, especially of perishable products, between the harvest and consumption steps of the chain [1,2]. The losses are mainly related to food spoilage caused by microbial contamination, molecules oxidation, and sensory characteristics deterioration [3,4]. These effects affect the safety of food products, threaten human health, and have a negative impact on consumer acceptance [4]. Considering the demands of both consumers and industry, edible coatings have been introduced as one alternative food packaging to replace plastic packaging and the synthetic preservatives traditionally incorporated to prolong the shelf life of different food products.

Edible coatings are thin layers applied directly on the food material’s surface. Food packaging is qualified as “edible” if it is an integral part of a food that may be eaten with it [5]. This material preserves and maximizes food quality, being widely used as a postharvest practice, especially in perishable products such as fruits and vegetables (F&V). Edible coatings protect food products from microbial contaminants, increase shelf life, decrease deterioration effects, and reduce lipid oxidation and moisture loss [1]. As with any other food-film production, edible coating formulation must consider different parameters, such as barrier properties (oxygen and carbon dioxide permeability), optical properties (they must be transparent and colorless), and sensory characteristics (they must be flavorless, tasteless, and odorless) [6]. Edible coatings can also enhance the sensory product attributes, like appearance and biochemical, physicochemical, and microbial stability. The nontoxicity and safety of this material and the low processing cost and the feasibility make edible coatings a good plastic-packaging alternative [5]. It must be noted that coatings do not always provide the proper attributes. Sometimes, the mechanical characteristics, poor transparency, or the lack of antimicrobial and oxidation resistance leads to the production of unsuitable films. Nevertheless, coatings are carriers for food additives like antimicrobials and antioxidants to improve both functional and physicochemical properties [3,4]. Moreover, edible coatings are considered environmentally friendly since they replace plastic packaging and reduce food waste by increasing the shelf-life storage of food products [1,6]. To produce edible coatings, various formulations can be used by engaging different structural compounds [6]. Thus, edible coatings can be classified into three groups, considering the nature of their elements: hydrocolloids (polysaccharides and comprising proteins), lipids, and blends of these compounds [7]. Furthermore, the usage of food industry by-products to produce biopolymers for edible coatings has already been considered [5]. This circular economy thinking strategy leads to reducing food waste, lowering the environmental impact of the food industry.

Fruits and vegetables (F&Vs) are products composed of vitamins, dietary fiber, phytochemicals, antioxidants, and minerals, whose consumption is linked to different health benefits such as the maintenance of human body immunity and the reduction in the risk of cardiovascular and cancer diseases, being fundamental for human nutrition [8,9,10,11]. F&Vs are widely consumed but problematic to manage along the supply chain since they are living tissues whose metabolic processes, such as CO_2_ production and O_2_ consumption during respiration, continue after harvest. Moreover, F&Vs have a high water content, so they are considered highly perishable products [8,9,10,11,12]. Postharvest deterioration can be minimized by controlling respiration rate, ethylene production, moisture loss, and microbial load. Both optimal storage conditions and postharvest technologies are needed to guarantee their storage stability and shelf-life extension [8]. F&Vs are products likely to be infected by Gram+ and Gram− bacteria, fungi, yeast, and molds because of the physiological and compositional changes occurring in the supply chain, making these products suitable substrates for microbial growth [10]. According to data, the main physico-chemical parameters affecting microbial spoilage of F&Vs are pH, temperature, and water activity (a_w_). Fruit pH is under 4.5, which promotes fungi growth (pH range between 3 and 8). Instead, vegetables have a pH range of between 4.8 and 6.5, allowing both fungi and bacteria growth [10]. The storage temperature recommended for F&Vs is between 0 and 5 °C since high temperatures accelerate the respiration process while low temperatures inhibit or delay microbial growth [10]. Nevertheless, psychotropic bacteria and fungi and chilling injuries must also be considered [10]. Finally, the liquid water available is a crucial factor for microbial growth (a_w_ between 0.97 and 1.00), even when harsh environmental conditions are applied. F&Vs have an a_w_ between 0.95 and 0.99 and are susceptible to microbial spoilage; however, reducing the water content is not an option [10].

F&V physicochemical characteristics lead to high product losses in the supply chain. Different studies have been carried out recently about food loss and waste (FLW). Food loss is the reduction in the quality or quantity of food that takes place in the chain, excluding retail, food service providers, and consumers, because of the decisions and actions of food suppliers. Food waste is the reduction in the quality or quantity of food resulting from decisions and actions by retailers, food services, and consumers [12]. According to different data, more than 20% of F&V production is lost or wasted [13], and 3–18% of the F&V loss occurs in the processing steps because of human errors, poor management, and technical failures [12]. In developing economies, food loss is linked to the postharvest and processing level, while developed economies are characterized by losing food at the retail and consumer level, being considered food waste [13]. Considering that the population is expected to reach 9.1 billion people by 2050, F&V stability and shelf-life extension have become issues for the food industry and society since increasing production cannot be the only solution to fulfill the demand [13]. Both stability and shelf life are linked to food quality and safety, being essential parameters for the food industry that affect the sensorial quality of the products [8]. Thus, food packaging is one key factor to prevent F&V waste, which takes place during the supply chain and once F&Vs are stored by consumers. Edible coatings may be a suitable alternative to traditional plastic food packaging since they can enhance the shelf life of F&Vs by reducing their respiration rate and loss of water and protecting them from physical damage and microbial spoilage, preventing postharvest loss [14]. In this review, the edible coating is presented as an alternative pathway to preserve fruits and vegetables, considering the environmental issues of current plastic packaging and the need for change in the food industry to decrease food waste. Moreover, a general vision of the present approaches is given, including the different procedures of edible coating production and the bioactive compounds incorporated into biopolymers used for this packaging.

## 2. Edible Coatings’ Contextualization

Edible coatings are biopolymer-based layers applied on the food surface that act as primary packaging and are considered a sustainable novel food packaging. Edible coatings are usually a mixture of film and additives [15]. Biopolymers are usually the main component of edible coatings and can be made from proteins, polysaccharides, or lipids. On the one hand, proteins and polysaccharides are excellent barriers against oxygen, lipids, and aromas but have moderate mechanical properties and high water permeability. On the other hand, lipids can be used as cohesive biomaterials; thanks to their characteristics, when transition temperature is achieved, they provide desirable gloss and an effective barrier for water loss [5]. To obtain a coating with better physicochemical properties, mixtures can be formulated. Moreover, different elements such as essential oils (EOs), bio-nanocomposites, and inorganic nanoparticles (NPs) can be incorporated into the mixture so the functional and physicochemical properties can be improved [5].

### 2.1. Film Formation and Application of Coat

There are different procedures for film formation, casting and extrusion being the methods most used. The casting method is a wet process used in the laboratory and in pilot scale. Casting is a three-step process: first, the biopolymer is solubilized in a solvent; then, the solution is cast in a mold; and finally, the cast solution is dried. Films formed by the casting method are characterized by their better particle–particle interaction, which leads to a more homogeneous film. This method has a low cost and does not need specialized equipment. Meanwhile, it has some drawbacks such as limited forms (only sheets and tubes are allowed), the potential trapping of toxic solvent inside the polymer, the protein denaturation because of the solvent, and the long time needed for the drying of the solvent [1]. On the other hand, the extrusion method, also known as the dry process, is a procedure that achieves better physicochemical properties and is widely used at industrial scale. It starts with the mixture of the film components and after it is compressed. The main advantages of this procedure are the short time and low energy consumption of the process, the better mechanical and optical properties of the film, the minimum usage of solvents, and the wide range of forms that can be obtained. Nevertheless, polymers must be of low moisture and tolerant to high temperatures for this method, and the cost and maintenance of the equipment are high [1]. Moreover, the application method of the coat directly affects the quality of the coating. Until now, dipping, spraying, and vacuum impregnation (Figure 1) are methods that have been developed.

*Dipping*: the product is immersed in the coating solution for 30 s–5 min, and then, the excess solution is drained. This method guarantees the application of the coat on all surfaces, even if they are rough [14].

*Spraying*: the droplet-form coating solution is homogeneously sprayed on the product surface [14]. Moreover, the solution surface is increased because of the droplets’ formation [1].

*Vacuum impregnation*: this method follows the dipping procedure but adds pressure, allowing the vacuum state [14].

### 2.2. Principle Macromolecules Used for the Edible Coating Formulation

#### 2.2.1. Polysaccharides

Polysaccharide-based coatings are characterized by their efficient O_2_ barrier because of their well-ordered hydrogen-bonded structure, their colorless and oil-free appearance, and their minimum caloric content, being suitable for the surface application on F&Vs. However, the moisture barrier capacity is limited by their hydrophilic nature [16], so blending with other compounds has also been considered. The polysaccharides most used are chitosan, starch, alginate, pectin, and cellulose, among others.

Chitosan (CH) is a co-polymer formed from deacetylated chitin in an alkaline medium. Chitin is a natural polymer of the exoskeleton structure of marine invertebrates, insects, fungi, algae, and yeast [17,18]. CH is characterized by its suitability for coat and film formation since it has permeable selectivity to CO_2_ and O_2_ and remarkable mechanical properties [17]. CH is a safe, natural allergen-free, and biocompatible polymer associated with different health benefits [16]. Moreover, CH coatings are nontoxic, biocompatible, and biodegradable and have antimicrobial effect over broad-spectrum pathogens, antioxidant activities [17], and excellent O_2_ and CO_2_ barrier properties [16]. CH film/coatings applied on the F&V surface positively impact their shelf life since decay-causing fungi, an antimicrobial effect, and an elicited host defense are produced [18]. Furthermore, CH’s semipermeable capacity can reduce the respiration rate by adjusting O_2_ permeability consumption and CO_2_ production, improving antioxidant activity [18]. Nevertheless, CH coatings have high water vapor permeability (WVP) [17], leading to a need to be fortified by incorporating other compounds. CH is also a biopolymer with great application for the food industry since it is a by-product produced during the processing of various exoskeleton crustaceans [19], so the use of these biopolymers for the production of edible coatings would support the circular economy system.

Starch is mainly formed by amylose and amylopectin, being suitable for coating production because of its mechanical and barrier properties [11]. Starch is a potential biopolymer for the food industry, not only for the suitable physicochemical properties but also for the sources in which it can be found, such as cassava, potatoes, sweet potatoes, corn, wheat, rice, and peas [11]. The hydrogen bond network of starch shows an excellent gas barrier for both CO_2_ and O_2_, as well as a low permeability of flavoring agents [11,16]. Moreover, starch biopolymers are odorless, tasteless, and colorless, suitable for the food industry [11]. Despite these suitable properties, starch biopolymers have high hydrophily [11], leading to an undesirable WVP that has to be improved by adding other molecules.

Alginate is a co-polymer extracted from seaweeds that forms a transparent, uniform, water-soluble, high-quality film/coat characterized by its colloidal characteristics, such as being a thickening, gel-forming, film-forming, and emulsion-stabilizing agent, as well as for its low permeability to lipophilic molecules and O_2_ [16,17]. This leads to the rear of the lipid oxidation processes of F&V and, consequently, to a reduction in weight loss and microbial growth [17]. Alginate-based coats also reduce shrinkage, moisture loss, oil absorption, and flavor and color loss [17].

Pectin is a generally recognized as safe (GRAS) molecule by the Food and Drug Administration (FDA). It is an amorphous, white-colored colloidal carbohydrate and the main plant cell wall component [6,20]. It is characterized by its suitable properties for the packaging sector of the food industry since it is nontoxic, biodegradable, and biocompatible and because of its permeability properties, gas barrier, and microbial controller abilities. Moreover, pectin has various technological properties, such as being an emulsifier, gelling agent, thickener, and stabilizer [6], and can be obtained from other food sector by-products, promoting a circular economy.

Cellulose-derivative-based coatings are characterized by their colorless and oil-free appearance and their minimal calories, while the moisture barrier is low because of their hydrophilic nature [16]. Carboxymethyl cellulose (CMC) is a cellulose derivative, an anionic linear long-chain water-soluble compound with high strength and structural integrity [21], suitable for the production of coatings applied to F&Vs. The presence of both hydroxyl and carboxylic groups in the CMC structure provides water binding and moisture sorption properties [21]. Moreover, CMC provides moisture, O_2_, CO_2_, aroma, and oil barrier improvement, as well as an increment of adhesion of the coating–surface interaction [21]. Finally, the antisenescence property of CMC reduces the ripening process in climacteric fruits [21], enhancing their shelf life.

#### 2.2.2. Proteins

Protein characteristics are suitable for edible film formation because of the amino acids’ position on the chain and the chain-to-chain interaction, which also determines the coating’s strength and gas and liquid permeability [16]. Protein-based edible coatings are considered excellent O_2_ blockers, even at low relative humidity, and from a nutritional point of view [16]. Different proteins have been applied in edible coating formation—soy, whey, gluten, and zein corn proteins being especially relevant in F&Vs. The food industry produces soy protein, zein protein, and gluten as by-products in activities such as soybean processing, defatting soybean favor, or production of wheat starch [19]. Moreover, casein, keratin, and gelatin are common protein residues from animal product processing [19]. Whey-protein-based coatings have accurate hindrance and excellent gas barrier properties at low relative humidity. Moreover, these coatings are suitable blockers of aroma compounds and oil, while their moisture barrier capacity is limited by their hydrophilic nature [16].

#### 2.2.3. Lipids

Since lipids are good barriers against water migration, lipid-based coatings are excellent moisture barriers. However, the high hydrophobicity of these coatings leads to extra-brittle and thicker properties, so a blending of lipids with proteins and polysaccharides is usually applied. Moreover, lipid-based coatings have been noticed with damaged appearance and gloss [16]. Paraffin-wax-based coatings are the main lipid-based films used as layers for F&V products. Films obtained with these waxes are characterized by their moisture barrier capacity and improvement of the outer surface appearance of different meals. Nevertheless, these layers are edible only when applied in a thin layer, while a thick layer must be disposed of before eating [16].

## 3. Food By-Products as Materials for Edible Coating Formation

Edible coating formulation requires at least one macromolecule to act as the biopolymer, which can be a polysaccharide, a protein, or a lipid. The blending of two or more compounds usually allows the obtaining of the best coat. With the application of these biopolymers on the surface of F&Vs, FLW is reduced and can be even more notable if the circular economy system is considered. The circular economy approach uses by-products produced by the food industry as sources of biopolymers used in edible coating formation. This policy is promoted by the European Union (EU), encouraging the food industry to upgrade the low-quality by-products obtained during processing to minimize waste. By-products are a source of bioactive compounds that can be introduced in edible coating formulation. If all or some of the coating components come from food industry by-products, the efficiency and sustainability of edible packaging will be guaranteed [19].

F&V postharvest processing also carries high amounts of by-products rich in compounds with suitable characteristics for edible coating formation [22]. These F&V by-products are known as plant-derived food by-products (PDFBPs) and are mainly seeds or kernels, pomace, peels, and leaves containing biopolymers such as cellulose, starch, pectin, and plant-based proteins, which are the main components of coatings and films [23]. Thus, wheat straws, wheat brans, and millet brans are a rich source of arabinoxylan, a hemicellulose polymer, while oat brans are rich in β-glucan. Moreover, fruit seeds or kernels are an excellent source of amylose, the main component of starch. CMC is found in rice stubbles and nanocellulose from wheat brans, obtaining nanofibers with high crystallinity and a large specific surface area rich in hydroxyl groups characterized by good biocompatibility and low cytotoxicity. Furthermore, apple pomace, mango, pineapple, and lime peels are rich in pectin, and lime peel pectin is a rapid-set gel former. Pineapple peel pectin incorporated into a commercial pectin film showed a higher water barrier property and antioxidant capacity. Finally, plant proteins can be extracted from different sources; coconut milk and rice bran are good examples [24].

Although plenty of data confirm the suitability of the by-product compounds generated by the food industry for edible coating formation, few studies have been developed using these molecules. In a study by Torres-León et al., the authors used mango peel flour and mango seed kernel to produce edible coatings applied to peach surfaces. For the mango peel flour coating, all compounds were used, while in the mango seed kernel, only the antioxidants were incorporated [25]. In another study by Grimaldi et al., all parts of onions, artichokes, and thistles were selected to incorporate into an edible coating since these vegetables entail a considerable amount of waste in the Italian food industry. Results showed that the coating had excellent mechanochemical properties [22]. Regarding food by-products, leaves are a big waste, so different studies have considered these vegetable parts as suitable matrices to obtain bioactive compounds. In 2022, Zhang et al. used loquat leaves to obtain an active extract to incorporate into the formulation of an edible coating applied to tangerines [26]. Aguilar-Veloz et al., Tesfay and Magwaza, and Chong and Brooks obtained leaf extracts of jackfruit, moringa, and haskap to formulate edible coatings to be applied to tomatoes, avocados, grape tomatoes, and bananas, respectively [27].

In summary, food by-products produced during food industry processing have different chemicals in their composition that can be used for edible coating and film production. Using peels, kernels, pomace, or crustaceous exoskeletons to recover polysaccharides and proteins incorporated as edible coatings leads to a promising circular economy achievement, as shown in Figure 2. However, because of the slight data available on studies that used food by-products for the edible coating formation and its potential exploitation (Table 1), further research is needed.

## 4. Improvement of the Physicochemical and Functional Characteristics of the Coatings

Biopolymers used as coats for food packaging are typically hydrocolloids, being alginates and chitosan that are broadly studied. Taking into account that biopolymers used as coats can be protein based or polysaccharide based and considering that both molecules are characterized by their hydrophilic capacity, the reinforcement of these compounds with other materials is convenient [50]. In this way, the physicochemical properties of these compounds can be improved by adding other elements, such as nanoparticles, essential oils, and bio-nanocomposites. Thus, the effects achieved with these components’ addition is described below and summarized in Figure 3.

### 4.1. Essential Oils

Essential oils (EOs) are natural derivative aromatic compounds that can be extracted from the seeds, stems, leaves, flowers, and fruits of plants [3]. Moreover, EOs are antimicrobial additives considered GRAS [14] and have attractive features for edible coating incorporation because of the preservative and antimicrobial ability of their compounds toward foodborne pathogens in materials such as vegetables, fruits, and meat products [14,51]. Since EOs are characterized by their hydrophobic substances, they are suitable for reducing vapor penetration by increasing the hydrophobicity of edible coatings [14,52]. EO features comprise the ability to increase the physical stability of active ingredients, the maintenance of aroma, taste, and flavor when added into nanoemulsions, and the increment of the effectiveness against foodborne pathogens [14]. Furthermore, EOs incorporated into a suitable delivery system are considered more efficient and protective to avoid dependency on other food components [14]. Nano-systems are commonly used as delivery systems of EOs to minimize the negative impact of these compounds in the sensory analysis of food products and to increase their stability in the food matrices [53]. Since EOs have antimicrobial and antioxidant properties, several studies have been developed about these volatile compounds, considering different matrices. Thus, cinnamon, oregano, and thyme are examples of matrices in which EOs have been studied with these activities [14,54].

### 4.2. Nanoemulsions

Nanoemulsions are oil-in-water or water-in-oil solutions that can be used for edible coating formulations to improve their physicochemical properties. Oil-in-water nanoemulsions are more suitable for this application since they can be easily fused with food-grade components, allowing a better scaling-up process [14]. The nanoemulsion mechanism is based on the nanodroplet formation of the emulsion covered by a film or a layer of a food ingredient. The particle size of these droplets (between 10 and 100 nm) increases the bioavailability and the chemical reactivity. Moreover, the functional qualities of the encapsulated component are improved because of the surface area increment [14,55], which is an essential factor for the promotion of the antibacterial EO characteristics and the improvement of the absorption of additional hydrophobic compounds [14,56]. These droplets are optically clear [14], and their bioavailability and chemical reactivity are higher because of the increment of the surface area of the particles [14]. The best procedure for the incorporation of EOs into edible coatings is in nanoemulsion, since this allows a minimum dose addition, leading to no adverse impact on sensory characteristics. Moreover, the nano-scale guarantees the effectiveness of shelf-life prolongation [14]. Oil-in-water nanoemulsions are considered the next-generation edible coatings since they can be fused with food-grade components, allowing more scope for the scaling-up step in the industry by applying a homogenization approach [14,57]. There are two processes for the nanoemulsion incorporation of edible coatings: a single-step nanoemulsion process in which all the ingredients are mixed in a coarse solution and then homogenized to create nanometric-sized droplets or a two-step nanoemulsion preparation where the aqueous solution is made first and, then, is combined with the biopolymer solution [58]. Nanoemulsions can be applied as edible coatings in different postharvest fruits such as papaya, mango, or strawberries [14].

### 4.3. Bio-Nanocomposites

Bio-nanocomposites are a mixture of various nanocomponents organized to create an obstacle against the entrance or existence of different molecules, such as oxygen or water vapor, reducing weight loss [59]. The concentration of the mixed materials influences the effect of the overall performance of polymer-based nanocomposites [50]. Moreover, it has been proved that the homogeneous dispersion of nanofillers leads to a better performance of bio-nanocomposites [50]. Nanocomposites also improve different properties of the biopolymers used for edible coating production. Among others, the mechanical properties, the barrier and thermal features, and the antimicrobial activity are improved [59]. Water barrier properties can be improved by adding Ag NPs, chitosan nanofibers, TiO_2_ NPs, Zein NPs, cellulose nanofillers, and copper oxide NPs, leading to lower solubility and an improved contact water angle. Moreover, NPs create a tortuous pathway by narrowing pore channels, which increases the diffusional path and reduces water diffusion [50]. In the same way, a uniform distribution of nanofillers in bio-nanocomposites leads to an excellent gas barrier by creating a complex pathway that restricts the movement of gas molecules through the material. These nanofillers’ distribution may change the interfacial region of the polymer matrix, improving gas permeability characteristics [50].

### 4.4. Inorganic Nanoparticles

Inorganic nanoparticles (NPs) are solid colloidal particles of 10–100 nm characterized by their stability, functionality, and biological activity [60]. NPs can load functional molecules, improving their stability and performance by encapsulation [60]. NP obtention methods are ionic crosslinking, covalent crosslinking, precipitation, and polyelectrolyte complexation [60]. These particles feature antimicrobial activity, so their incorporation has a protective function [15,61]. Moreover, the increased surface area leads to a better reinforcement effect in the matrix [59]. Thus, using NPs in the food industry relies on their ability to be easily dispersed into matrices, minimizing the adverse flavor impact, while the diffusion and bioavailability are enhanced [61]. In this way, incorporating NPs into edible coatings can be a suitable option to reduce the adverse effects of antimicrobial substances on food odors [61].

## 5. Application of Edible Coatings on Fruits and Vegetables

### 5.1. Current Application of Edible Coatings Applied on Fruits and Vegetables

Nowadays, edible coatings are studied for perishable food products, such as F&Vs, meat and poultry, and fresh fish. On the one hand, microbial contamination is common in these products because of their high water content. F&Vs are prone to water loss, mechanical damage, and sensory changes during storage, leading to economic loss [3]. Mechanical injury, environmental stress, and pathological breakdown are also situations that reduce these perishable food products’ shelf life [59]. These effects also impact the consumers’ acceptance and health [3]. Different studies have applied biopolymers with bioactive compounds (such as EOs), inorganic NPs, or bio-nanocomposites as edible coatings to prolong these products’ shelf life. In this section, the current application of edible coatings (Table 2) is discussed, providing several examples where F&V shelf life is improved (Table 1).

Several studies have been carried out using edible coatings as shelf-life extenders. Polysaccharides are the most common macromolecules used in EC production. Sucheta et al. studied the tomato’s changes during 30 days of storage at 25 °C using a pectin-based edible coating mixed with corn flour and beetroot powder in different proportions. Results showed how the coating produced with 50% pectin and 50% corn flour (PCF) achieved the best weight loss, decay percentage, respiration rate, and biochemical quality. Moreover, PCF and P (100% pectin) coating could maintain maximum glossiness and minimum shrinkage of the tomato’s pericarp without causing off-flavors [30]. Rodriguez-Garcia et al. also studied the tomato changes that took place during 12 days of storage at 25 °C. The edible coating was prepared using citrus peel pectin as the main macromolecule, mixed with oregano EO. Results showed the antifungal effect of the EC on inoculated tomatoes because of the EO addition. Moreover, the total phenol content and the antioxidant activity were increased without affecting the sensory acceptability of the tomatoes [31]. Another study explored the tomato’s variations after being coated with alginate mixed with aloe vera, and ZnO-NPs were measured during 16 days of storage at room temperature. Results showed the applicability of this edible coating since the UV shielding and water barrier and the thermal, mechanical, and antimicrobial properties were excellent. The authors demonstrated that the improved properties resulted from the synergic action of the alginate mixed with ZnO-NPs and aloe vera [36]. Strawberries were also among the focus fruits for the shelf-life extension. Khodaei et al. applied an edible coating made of CMC, Persian gum (PG), low methoxyl pectin (LMP), or tragacanth gum (TG) in strawberries stored at fridge temperature (4 °C) for 16 days. To analyze the effect of the different coating treatments on the strawberries’ shelf-life, the TOPSIS method was applied, showing that the CMC coating has the best results in reducing weight loss and spoilage and preserving nutritional ingredients [33]. The shelf-life prolongation of coated strawberries was also studied. The coatings were made of pullulan (a water-soluble polysaccharide), and cinnamon EO was incorporated in a nanoemulsion structure. The edible coating addition led to different improvements in the mass loss delay, firmness, total soluble solids, and titratable acidity during the 6 days of storage at room temperature [39]. In the same way as strawberries, cherry fruits are highly perishable. Sweet cherries were coated with a polysaccharide-based coat of alginate and chitosan and mixed with olive leaf extract. After 20 days of storage at 25 °C, coated cherries showed retardation of the ripening process and maximum retention of phenolic compounds. Furthermore, the authors determined the correlation between antioxidant activity and the retention of phenolic compounds [28]. Coated grapes have been studied using a polysaccharide-based coating of chitosan (1%) and a combination of chitosan (1%) and gum ghatti (1%) applied to the grapes’ surface before their storage at fridge temperature (1 °C) for 60 days. After comparing the results, the chitosan and gum ghatti showed better antifungal activity. This coat also gave good results regarding nutritional properties, phenolic compounds, and antioxidant activity maintenance [35].

Regarding protein-based coatings, fewer studies have been conducted. Protein-based coatings mixed with transglutaminase were applied to apples, potatoes, and carrots cut in slices and stored at fresh temperatures for 10 days. In this study, positive results were shown when the transglutaminase was incorporated into the edible coating, reducing weight loss, preventing microbial growth, and maintaining antioxidant activity during the 10-day storage. Furthermore, apples, carrots, and potatoes showed no significant changes in their hardness and chewiness [32]. Another study compared the addition of lemon oil, lemongrass EO, and non-incorporated oils in whey-protein edible coatings applied onto the surface of pears. Results showed how the coating with lemon oil (1%) and lemongrass EO (0.5%) reduced weight loss, WVP, oxygen, and carbon dioxide. However, those pears coated with lemon oil had a reduction in firmness after 28 days at fresh storage with 80% humidity, while those coated with lemongrass EO showed higher brownness because of the nature of the oil and its natural yellow color. Regarding acceptability, both coatings showed a slight reduction in acceptability compared with the EC without the oils [34]. Walnut flour protein was applied to the kernel surface of walnuts to improve their storage for 40 days at 84 °C. Edible coatings preserved the sensory characteristics, especially those regarding lipid deterioration (e.g., lipidic peroxidation that leads to rancid flavor) [41].

According to Table 1, EOs are commonly chosen to improve the features of the coatings. Olive leaf extract, orange, oregano, thyme, cinnamon, and lemon EOs are broadly incorporated into edible coatings. Reduction in water loss and hardness, fewer color changes, and lower respiration rates are the most common advantages when edible coatings formulated with EOs are applied on F&V products. Moreover, storage time is highly incremented when the edible coating is applied on fruits, especially suitable on those fruits for which a long period passes between harvest and selling. Therefore, the edible coating incorporation on F&V surfaces improved the shelf-life extension of these perishable products, which may lead to lower waste production and, finally, to fewer economic losses.

### 5.2. Effects of Edible Coatings in the Sensory Characteristics

Edible coatings seem a suitable pathway for the shelf-life extension of F&Vs, producing some improvements such as retardation of maturation, inhibition of enzymatic brown reactions, or reduction in respiration rates, among others. However, the impact of the edible coating application on the acceptance of the product, depending on firmness, color, flavor, taste, smell, hardness, and overall acceptability, should be considered. Basaglia et al. studied the changes in color and firmness of coated pineapples, and results showed no significant variations in color until day 7, with a lower decrease in brightness compared to uncoated pineapples. No significant changes were measured in firmness until day 9. Moreover, 12 trained judges analyzed aroma and overall evaluation, finding no significant difference until day 5 [37]. Manzoor et al. determined the firmness of fresh-cut kiwifruit coated with sodium alginate, ascorbic acid, and vanillin, showing slower changes in those coated slices than in the uncoated ones [40]. However, Tabassum et al. determined the changes in color and firmness when guava leaf and lemon extract were applied to bananas using a scale ranging from 1 to 7 and 1 to 5, respectively. Results showed that coated bananas maintained firmness and color for 2 and 5 days more than those uncoated, respectively [29]. In a different study, 15 semi-trained judges evaluated coated tomatoes’ taste, firmness, and visual appearance. After a 15-day storage period, the acceptance rate was higher than for the uncoated. Nevertheless, a distinct but pleasant difference in the coated tomatoes was detected. Regarding firmness and color, results showed higher visual brightness of coated tomatoes compared with those uncoated. Finally, firmness of coated tomatoes was reduced by 28%, while uncoated tomatoes showed a reduction in firmness of 61% [38]. The sensory changes of tomatoes coated with whey protein, xanthan gum, and clove oil were also analyzed by a trained panelist. Color, texture, taste, flavor, and overall acceptability were evaluated. After 15 days of storage, the coated tomatoes were found acceptable, whereas the uncoated tomatoes showed a desiccated appearance. The coated tomatoes showed no adverse effect on color, texture, taste, and flavor [44]. Regarding the data available considering sensory characteristics of the F&Vs coated, positive results are found. Overall, there is an improvement in the firmness, color, flavor, smell, and taste of the samples coated, guaranteeing the consumers’ acceptability of F&Vs.

### 5.3. Edible Coatings Regulation

When EC are added onto F&V surfaces, the materials of these coatings are in direct contact with the food product. Therefore, the safety approval of the corresponding authorities is necessary for the commercialization of these products [65]. In this way, the Food and Drug Administration (FDA) and the European Food Safety Authority (EFSA) have developed codes regarding the proper application of packaging and food contact materials (FCM) [65], guaranteeing the consumer’s health [66]. Thus, three different categories have been described: FCM, defined as the materials and articles that are in direct contact with food, such as nanoparticles and antimicrobial and antioxidant compounds, that in the case of the European Union are legislated by Regulation (EC) 1935/2004; food contact substances (FCS), which are the components of the main material, and food contact articles (FCA), which are the final product (whether they are coatings or films) [65,66]. It is important to remark that the edible coatings added to the surfaces of F&Vs must be a recognized GRAS or food additive by the FDA, and the main compounds used in the production of EC are generally in the additives list made by the European authorities included in Regulation (EC) 1333/2008 [65].

In conclusion, when edible coatings are developed, it is important to use compounds that are considered safe or are included in the additive lists of the different authorities so consumer health is not compromised. In this way, it would be interesting for the European authorities to develop a more specific legislation for EC, since it seems to be an alternative with potential for the improvement of the shelf life and quality of perishable products such as F&Vs.

## 6. Conclusions

Edible coatings are a suitable alternative to traditional plastic packaging that is being widely studied nowadays, especially for perishable products such as fruits and vegetables. The need for food waste reduction, the consumers’ interest in incorporating more fresh products in their diet, and the awareness of society about the environment are key points and the main reasons why edible coatings are receiving so much interest from the scientific community. Different formulations can be used to achieve the best edible coating properties for one specific product. Furthermore, edible coatings are suitable for incorporating substances that provide added value to the product, such as bioactive compounds, essential oils, or nanoparticles, and in many cases, these compounds can be obtained from food industry by-products, contributing to the circular economy. This review aimed to provide data about edible coatings, their formulation, and their application procedures and to give an approach to studies considering edible coatings applied to fruits and vegetables. The data collected about edible coatings show that this pathway should be considered by the food industry, not only for the advantages from the point of view of the prolongation of perishable products’ shelf-life but for the possibility of using other food by-products, such as pectin and chitosan, as biopolymers for coating formulations. Since inorganic nanoparticles and other elements can be incorporated into edible coatings, further investigations considering toxicity must be performed to guarantee consumers’ health.

## Figures and Tables

**Figure 1 foods-12-03570-f001:**
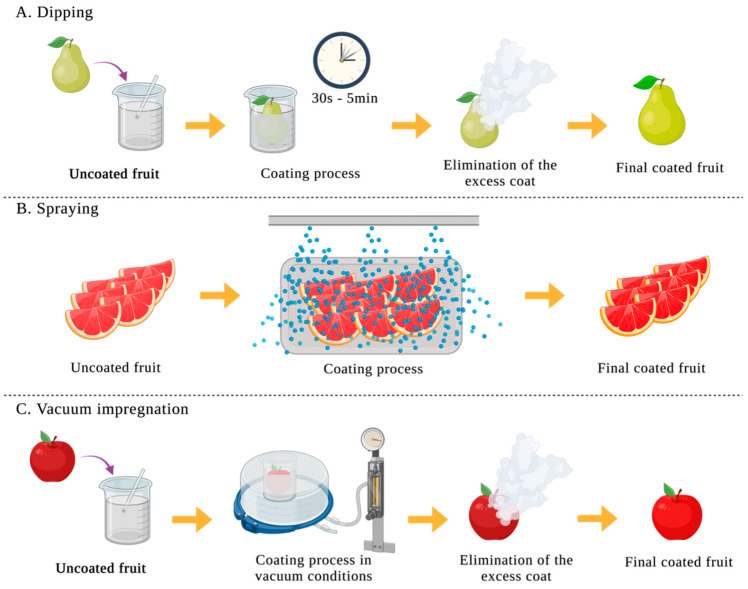
Schematic representation of the processes used for the coating application on fruits and vegetables. (**A**): dipping; (**B**): spraying; (**C**): vacuum impregnation.

**Figure 2 foods-12-03570-f002:**
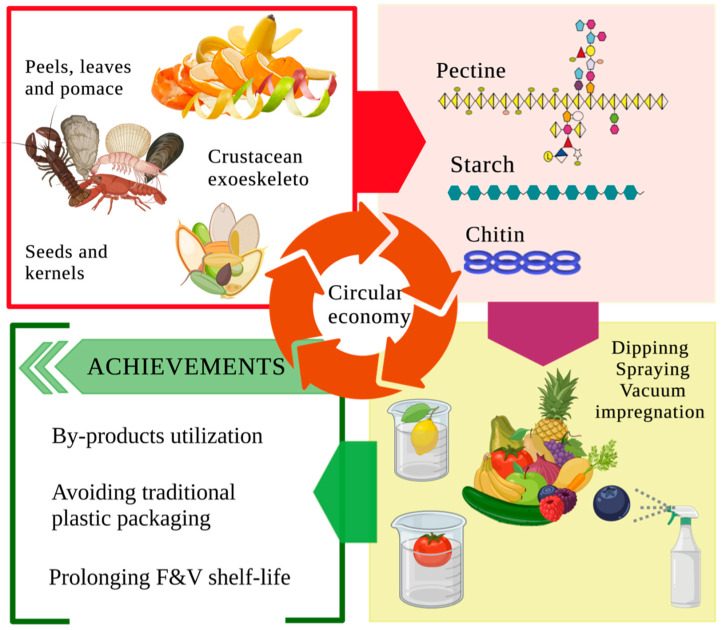
Application of food industry by-products as potential components to be used as edible coatings and films in fruits and vegetables.

**Figure 3 foods-12-03570-f003:**
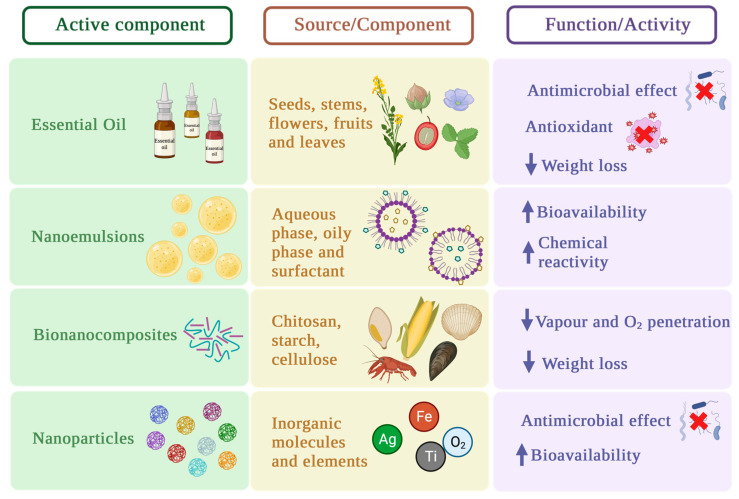
Principal components used for the improvement of edible coatings and their function.

**Table 1 foods-12-03570-t001:** Improvements of different food products after their storage with edible coatings.

Edible Coating	Macro-Molecule	Remarkable Component	Food Product	Improvement	T (°C)	t (Days)	RH (%)	Ref.
Alginate (3%) + Chitosan (1%) + Olive leaf extract	PS	Olive leaf extract	Cherry fruits	Retardation of maturation, Anthocyanin incrementation	25	20	65	[28]
Guava leaf extract (20%) + Lemon extract (15%)	-	Guava leaf extract + Lemon extract	Banana	Reduction in color changes, Preservation of vitamin C	NS	14	NS	[29]
Commercial pectin + corn-flour starch + beetroot powder	PS	Corn-flour starch//Corn-flour–beetroot powder	Tomatoes	Lower weight loss, Lower decay percentage, Lower respiration rate	25	30	80–85	[30]
Pectin + Oregano essential oil (36.1 mg/mL)	PS	Oregano EO	Tomatoes	Lower fungal decay, Increase in antioxidant activity	25	12	NS	[31]
Whey protein pectin + pectin + transglutaminase	PR and PS	Transglutaminase	Apples, Potatoes, Carrots	Lower weight loss, Inhibition of microbial growth, Antioxidant activity preservation	4–6	10	NS	[32]
Glycerol + Tragacanth gum (0.6%)	PS	Tragacanth gum	Strawberries	Reduction in the rate of deterioration in ascorbic acid, total phenolics and anthocyanins	4	16	NS	[33]
Glycerol + CMC (1%)		CMC						
Glycerol +LMP (2%)		LMP						
Glycerol + PG (4%)		PG						
Whey protein (8%) + lemon oil (1%)	PR	Lemon oil	Pears	Reduction in color changes, Reduction in loss of hardness, Reduction in loss of polyphenols and flavonoids	4	7–28	80	[34]
Whey protein (8%) + lemon grass essential oil (0.5%)	PR	Lemongrass EO						
Chitosan (1%) + Gum ghatti (1%)	PS	Gum ghatti	Grapes	Retention of phenolic acids content, Reduction in yeast-mold growth	1	60	85–90	[35]
Alginate + Aloe vera + ZnO-NPs	PS	Aloe vera ZnO-NPs	Tomatoes	No spoilage during the storage	RT	16	NS	[36]
PET + Chitosan (2%) + Cinnamon essential oil (0.5%)	PS	Cinnamon EO	Pineapple	Lower weight loss, Lower decrease in L*, Microbial growth retardation	5	15	NS	[37]
Sodium alginate + sweet orange essential oil (5%)	PS	Sweet orange EO	Tomatoes	Eradication of sessile and planktonic forms of *Salmonella* and *Listeria*, Lower weight loss	22	15	NS	[38]
Pullulan + cinnamon EO	PS	Cinnamon EO	Strawberry	Delay in mass loss, decay percentage, and firmness	20	6	70–75%	[39]
Sodium alginate (2%) + ascorbic acid (0.5%) + vanillin (1%)	PS	Ascorbic acid, Vanillin	Kiwifruit	Lower decay, Lower ascorbic acid loss	5	7	NS	[40]
Sodium alginate (2%) + ascorbic acid (0.5%) + vanillin (0.5%)				Lower weight loss				
Walnut flour protein	PR	-	Walnuts	Protection against lipid deterioration, Preservation of the sensory characteristics	40	84	NS	[41]
Soy protein + ferulic acid	PR	Ferulic acid	Fresh-cut apple	Weight loss control, Firmness control	10	7	50	[42]
Soy protein + cysteine (1%)	PR	Cysteine	Fresh-cut eggplant	Enzymatic browning control	8–9	5	NS	[43]
Whey protein +. Xanthan gum + Clove oil	PR	Clove oil	Tomatoes	Improvements in firmness and color, Respiration is inhibited	15	20	85	[44]
Whey protein nanofibrils + glycerol + trehalose	PR	Glycerol and trehalose	Fresh-cut apple	Enzymatic browning control	10	4	NS	[45]
Whey protein nanofibrils		-		Hydrophobic and antioxidant activity				
Aloe vera 50% gel	Gel	Aloe vera	Papaya fruit	Control disease pathogens, Delay ripening, Water loss control, Respiration rate reduction	15	28	68–70	[46]
Chitosan-gelatin	PS	-	Red bell peppers	Microbial spoilage reduction, Respiration rate maintenance, Nutritional content maintenance	7; 20	14	-	[47]
					21	7		
					14	20		
Chitosan + HPMC + bergamot EO	PS	Bergamot EO	Grapes	Weight loss reduction, Respiration rate reduction, Firmness improvement, Antimicrobial effect	1–2	22	-	[48]
Pea starch + guar gum	PS	-	Oranges	Shelf-life extension, Higher perception of off flavors, Better sensory scores	20	7	90–95	[49]

**Abbreviations:** T: temperature; t: time; RH: relative humidity; EO essential oil; RT room temperature; NS not specified; CMC methyl cellulose; LMP: Low methoxyl pectin; PG: Persian gum; PS: polysaccharide; PR: protein; L: lipids.

**Table 2 foods-12-03570-t002:** Application of by-product compounds in edible coating production.

Product	Subproduct	Extraction	Conditions	Compound	Formulation	Application	Product	Ref.
Mango	Mango peel flour	Drying pretreatment	60 °C; 48 h	-	1.09% MPF 0.33% Gly	Casting	Peach	[25]
	Mango seed kernel	Solvent extraction	EtOH 90%; 75 °C	Antioxidant compounds	1.09% MPF 0.33% Gly 0.078 g/L EMS			
Onion	Leaves, stems, flowers	UAE	Acetone	Phenolic compounds	1% SA 0.3 g/g Gly/SA 0.04 g/g CaCO_3_/SA 5.4 g/g GDL/CaCO_3_	-	-	[22]
Artichoke								
Thistle								
Mango	Kernel	Solvent extraction	Sodium metabisulphite 0.16%	Amylose; amylopectin	Gly:Sorbitol 1:1 Starch 50%	Dipping	Tomato	[62]
Olive leaf	Leaves	Solvent extraction	EtOH 40%; 60 °C; 120 min	Olive leaf extract	3% SA 10% Gly 2% CaCl_2_ 0.01 g/mL Chitosan 0.02 g/mL OLE	Dipping	Sweet cherries	[28]
Loquat leaf	Leaves	Reflux extraction	EtOH 50%; 196 °C	Loquat leaf extract	10 g/L SA 0.7 g/L CA 10 g/L SE 0.5 g/L AsA 0.5 g/L PS	Dipping	Tangerines	[26]
Jackfruit leaf	Leaves	MAE	EtOH:H_2_O 4:1; 840 W; 2 min	Jackfruit leaf extract	1.5% pectin (*w*/*v*) 30% (*w*/*w*) Gly 10% (*w*/*w*) Beeswax JLE 5 mg/mL	Wounded	Tomatoes	[27]
Moringa leaf	Leaves	NS	-	Moringa leaf extract	MLE 2% Chitosan 0.5% CMC 0.5%	Dipping	Avocado	[63]
Haskap leaf	Leaves	ATPE	Sodium phosphate 10%, EtOH 37%, H_2_O 53%; 5 min	Haskap leaf bioactive compounds	PPC 10% 0.91% Gly 10% SPHLE/ASHLE 1.7 g CA/MA	Filming	Grape tomatoes	[64]
						Dipping	Bananas	

**Abbreviations:** UAE: ultrasound-assisted extraction; MAE: microwave-assisted extraction; ATPE: aqueous two-phase extraction; MPF: mango peel flour; Gly: glycerol; EMS: extract mango seed; SA: sodium alginate; GDL: D-(+)-gluconic acid-σ-lactone; OLE: olive leaf extract; JLE: jackfruit leaf extract; CA: citric acid; SE: sucrose ester; AsA: ascorbic acid; PS: potassium sorbate; PPC: pea protein concentrate; SPHLE: sodium phosphate haskap leaf extract; ASHLE: ammonium sulphate haskap leaf extract; MA: malic acid; NS: not specified.

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
