# Peer review of "Edible Coatings as a Natural Packaging System to Improve Fruit and Vegetable Shelf Life and Quality"

_foods, 2023, doi:10.3390/foods12193570_

Round 1

Reviewer 1 Report

The review manuscript was written with 64 articles. 95% of articles are published in the last 10 years, of which 78% in the last 5 years. The topic is very topical and covers a very large number of publications in the scientific and professional literature. The authors omitted articles in which coatings were made, for example, on the basis of pullulan (except for one).

Comments details:

Lines 65-67: What's the rationale for listing chitosan here? It's not the only polysaccharide used as a base for edible coatings.

Lines 167-168: Other polysaccharides omitted, e.g. pullulan, which is a polysaccharide that is very often used as an edible coating.

Lines 220-233: This section (2.2.2. Proteins) only lists the proteins and does not describe their uses. Consistently it should be as in section 2.2.1. Polysaccharides, which describes chitosan, starch, alginate, pectin and cellulose.

Lines 234-243: Note as above.

Lines 295-296, Table 1: Latin names (Salmonella and Listeria) should be written in italics. The "Active component" column is debatable. This column lists the OEs and base coating substances, e.g. Corn-flour starch // Corn-flower-beetroot powder, Tragacanth gum, CMC. Are the base substances active ingredients? The coating can work as a whole, the active component is e.g. essential oils.

Lines 280-381, Fig.3: Correct the spelling of "cellulose".

Author Response

Lines 65-67: What's the rationale for listing chitosan here? It's not the only polysaccharide used as a base for edible coatings.

Response: Thank you for your comment. The authors have re-read the paragraph and decided to eliminate those lines.

Lines 167-168: Other polysaccharides omitted, e.g., pullulan, which is a polysaccharide that is very often used as an edible coating.

Response: The authors appreciate your comment. The polysaccharides mentioned in these lines are the ones described below, so we did not mention pullulan. However, in the enumeration list authors included the words “among others” to clarify that there are more polysaccharides used as edible coatings such as pullulan.

Lines 220-233: This section (2.2.2. Proteins) only lists the proteins and does not describe their uses. Consistently it should be as in section 2.2.1. Polysaccharides, which describes chitosan, starch, alginate, pectin, and cellulose.

Lines 234-243: Note as above.

Response: The authors agree with your comment. It is true that this section lacks description and the same happens for lipids. However, there is not as much information for proteins and lipids compared to polysaccharides, so the discussion could not be performed in a similar way as section 2.2.1 Polysaccharides. Anyhow, we did not want to leave these molecules behind because of their importance and we decided to include them.

Lines 295-296, Table 1: Latin names (Salmonella and Listeria) should be written in italics. The "Active component" column is debatable. This column lists the OEs and base coating substances, e.g. Corn-flour starch // Corn-flower-beetroot powder, Tragacanth gum, CMC. Are the base substances active ingredients? The coating can work as a whole, the active component is e.g. essential oils.

Response: Thank you for your comment. The Latin names of Salmonella and Listeria have been corrected and written in italics. Moreover, as pointed out, the term “active component” was not accurate for all the rows, so we have replaced it with “remarkable component”.

Lines 280-381, Fig.3: Correct the spelling of "cellulose"

Response: Thank you for noticing. The word “cellulose” has been corrected in Fig 3.

Reviewer 2 Report

This is a topic of great interest. In literature, various articles, reports and reviews explore this topic in various parts. Therfore, it is very difficult not to be repetitive. This review explores key topics in some parts that are in common with other research articles and reviews, but it was interesting to see that the authors brought the concep of circular economy and the use of by-products into this discussion.

Detailed comments

The title is a bit misleading, it cannot be understood as an alternative to traditional packaging. But a natural system that can help to maintain the quality of fruits and vegetables.

Lines 47-50: It is repetitive, the concept has already been said previously

Lines 56-58: It is repetitive, the concept has already been said previously

Line 122-127: Rewrite it is too long

In general the Emglish is readable, only minor changes are required

Author Response

The title is a bit misleading, it cannot be understood as an alternative to traditional packaging. but a natural system that can help to maintain the quality of fruits and vegetables.

Response: Thank you very much for your appreciation. The title has already adapted.

“Edible coatings as a natural packaging system to improve fruits and vegetables shelf-life and quality”.

Lines 47-50: It is repetitive, the concept has already been said previously

Response: Thank you very much for your appreciation. The sentence has been deleted as requested.

Lines 56-58: It is repetitive, the concept has already been said previously

Response: Thank you very much for your appreciation. The authors have thoroughly revised the manuscript and have decided to maintain the sentence but shorten it.

Nevertheless, coatings are carriers for food additives like antimicrobials and antioxidants to improve both functional and physicochemical properties [3,4].

Line 122-127: Rewrite it is too long

Response: Thank you for your suggestion. The authors have rewritten this part so it can be easily read.

“Biopolymers are usually the main component of edible coatings and can be made of proteins, polysaccharides, or lipids. On the one hand, proteins, and polysaccharides are excellent barriers against oxygen, lipids, and aromas but have moderate mechanical properties and high water permeability. On the other hand, lipids can be used as cohesive biomaterials thanks to their characteristics when transition temperature is achieved, they provide desirable gloss and an effective barrier for water loss”.